# Forest Dependence of Rural Communities in the Republic of Moldova

Nicolae Talpă [1,*], Aurel Lozan [2], Aureliu Florin Hălălișan [1] and Bogdan Popa [1]

1   Department of Forest Engineering, Faculty of Silviculture and Forest Engineering, Transilvania University of Brașov, Șirul Beethoven No. 1, 500123 Brașov, Romania; aureliu.halalisan@unitbv.ro (A.F.H.); popa.bogdan@unitbv.ro (B.P.)
2   Biodiversity and Forest Ecology Expert, MD-2062 Chișinău, Moldova; protectingnature@gmail.com
*   Correspondence: nicolae.talpa@unitbv.ro; Tel.: +40-268-413-000

**Abstract:** The high dependency on forest resources and the fact that forests play an important role in the lives of people in poorer rural areas are well known forest characteristics. This depicts a deep connection between people and nature. For the rural communities, forest ecosystems display another important role, namely in alleviating poverty through stable provisions of vital functions and livelihoods. The present study aims to identify what influences the current level of the local communities of the Republic of Moldova's dependence on forests, who still face poverty-related challenges, and how ecosystem services provided by forests are perceived by the rural population. After six years since the last forest dependency research, this time the level of dependence is investigated using the same methodology, but through improved socio-economic conditions. Although the consumption of non-timber forest products decreased, the pressure on forest resources remained at the same level due to the high dependence of communities on firewood. The highest dependency was found in low-income households, manifested by their necessity to spend an average share of 18.8% from their total income on firewood due to their restricted access to forest resources. Since most Moldovans rely more heavily on subsistence-oriented forest products such as fuelwood, forest management sustainability efforts might not be achieved as long as the need for wood products exceeds the supply, and neither will the living conditions of the poor be improved. Solutions should be sought based on cross-sectoral and long-term approaches by involving all stakeholders, and not neglecting local communities.

**Keywords:** Moldova; households; local communities; forest dependency; socio-economic factors

## 1. Introduction

Forests play an important role in the lives of rural populations through the supply of crucial provisioning, regulating, supporting, and cultural services [1]. There are multiple reasons why natural resource management policies should consider what forest ecosystems offer to local communities, and how this makes a significant impact on strategy and policy development [2,3]. By meeting needs or providing a potential source of income from the use of forest resources, forest ecosystems can make a significant contribution to poverty alleviation [4,5]. Specifically, forest management policies should be targeted to meet subsistence needs and encourage those who are able to increase their income through forestry activities [6] or who directly depend on access to the forest resource for basic needs [7]. However, unsustainable forest practices often tend to work against the interests of the poor [8,9], sometimes resulting in transfers that favor the richest [6,10]. Uncovering where the intention to conserve forests and meet people's increased demand for forest resources are interlinked is another angle of the story [4,11], one which could allow rural people and forests to coexist in a win–win relationship [12,13]. Externally controlled conservation initiatives, including the expansion of protected area networks, often give rise to considerable 'human–nature' conflicts as they involve strategies that may

change local practices and threaten social outcomes, thus posing risks which may render conservation efforts ineffective [14]. The concept of Ecosystem Services (ES) [15] is based on the interdependence between natural and human well-being [16,17], and for this to be true, local people should be the primary stakeholders in designing forest resource management policies while their lives remain connected with forests [18,19]. The interaction that takes place between local communities and forest ecosystems requires a thorough analysis that considers all hidden facets [20], and for the needs of forest-dependent communities to be met in a sustainable way this must be the primary focus of forest management [18,21].

To some extent, all people are dependent on the forest, which combines both historical and modern values defined as human–forest relationships [22], with some groups being more dependent on this resource from the perspective of meeting basic human needs [3]. Forests have significant potential to improve living conditions, especially for rural people [4,23]. This has also been demonstrated by the Poverty Environment Network's (PEN) project, one of the largest quantitative research projects on forests and rural livelihoods. The project was coordinated by the Centre for International Forestry Research (CIFOR) and employed a method that involved quantitative surveys of community members in rural areas regarding their households' wealth and sources of income for families. In addition to providing opportunities for global comparisons, this unique method yielded results that demonstrated two central elements in all the cases investigated: (1) wealthier (higher-income) rural households use higher amounts of forest products [10,24] and (2) poorer households are more dependent on forest resources through their higher share of total household income [2,24]. This proves how important forests are for rural incomes and how dependent rural households can be on the resources (sometimes at subsistence level) provided by forest ecosystems, although these resources may differ from one country or region to another. The information included in the PEN project describes the current situation and demonstrates not only the role of forest and environmental resources for rural households, but also how important it is to produce tools that can help design appropriate policies prioritizing forests' contribution toward satisfying people's essential needs as well as ensuring these resources are continuously generated by sustainably managed ecosystems [25].

Most of the PEN project research has been conducted in poor and developing tropical and subtropical countries located in Latin America, Asia, and Sub-Saharan Africa [2], where forest management is often poorly carried out or non-existent [26]; for these areas, the role of forest products in the lives of rural households is continuously changing as their standard of living increases. The method designed by CIFOR has also been replicated without significant methodological changes in the EU-funded regional European Neighborhood and Partnership Instrument East Countries Forest Law Enforcement and Governance II (ENPI FLEG II) program that involved seven countries (Armenia, Azerbaijan, Belarus, Georgia, Moldova, Ukraine, and Russia). Analysis in the former Soviet space [25] has shown that income from forest resources has a significant share in the total income of the poor, thus expressing their dependence on the forest, and at the same time, that the richest have higher incomes from the use of forest ecosystems. The majority of communities signal that forest products are becoming less available by pointing towards the continuous decline in provisional services generated by forest ecosystems, with the main drivers of this being over-harvesting, illegal logging, and climate change [25]. Similar to PEN studies in tropical and subtropical countries [2], the study in the ex-soviet space illustrates how important forest resources are to rural households and highlights the main principles that can be used for developing appropriate policies that should consider the needs of rural communities [25]. Increasing total population income and alleviating poverty will not reduce pressure on natural resources; likewise, limiting access to natural resources through exclusively conservationist policies will only jeopardize the living standards of poorer households.

The human–forest relationships in the poorly forested Moldova, combining historical and recent aspects, should be regarded in the context of its rapidly changing socio-economic

developments. Almost six years after the first forest dependency analysis conducted in Moldova based on the CIFOR methodology [27], under FLEG project phase II, the present study aims to identify, based on the same methodology, how important forest ES can be for rural communities and how much the role of forests is currently influenced by various factors.

## 2. National Context

The forest-related legislation in Moldova is not as imperfect compared as how it is perceived [28,29], but rather its implementation raises questions over forest conditions and planned management [30,31]. Forests' quantitative indicators are generally characterized by low values, with a per capita total standing wood volume of 11.3 m$^3$ and a per capita harvested wood volume of nearly 0.16 m$^3$/year [32]. Revenues from the forestry activity within the Agency Moldsilva (Moldsilva)—which is the central authority for the forest and hunting state policy [33] and the umbrella coordinator of a network of state forest management entities [34]—come mainly from wood/timber harvesting [32,35] through a self-financing mechanism introduced since 1998 which is considered to induce an increased pressure on existing forests [31,36].

Non-timber forest products (NTFPs) harvested by Moldsilva's network of state forest entities are insignificant and vary depending on environmental factors and market demands [37], while their annual trade activity provides an average cash income of only 2–4% from the total turnover which relies on wood sales [38]. Moldsilva's NTFPs production is not well developed economically (new markets and obsolete processing) nor ecologically (species diversification and innovation) and lies mainly with culinary nut-fruit species. On the one hand, this low-income value from the valorization of NFTPs is influenced by two linked factors, an increased land degradation and a superfluous transfer/change of land use categories, so currently there are only 289.2 ha of croplands under cultivation of various fruit-bearing forest species [39]. On the other hand, the lack of specialization (i.e., specialist units for collection and processing of the raw material) and an insufficient investment mechanism along with a low interest in promoting NTFPs are the main obstacles of this important sub-sectoral development [34,40].

The inefficiency of the current use practices of lands intended for forest fruit/berry production along with the lack of specialized personnel and unclear long-term financial perspectives or trade marketing vision [40] will most likely not bring any encouraging change in the near future. The trend in NTFPs production and processing will remain within the same figures as it is now [41], thus supporting the assumption that both production and commercialization revenues will not increase considerably. Moldsilva states that to improve the NTFPs sector and make real progress toward novel processing and marketing, it is necessary to develop a proper regulatory framework that will enable more effective management approaches including carrying out marketing research, build up the capacity of existing administrative structures, increase personnel training, and create more favorable conditions for investments into both infrastructure and technology [41]. This means that an analysis of the real potential of domestic forestlands for developing the NTFPs sector based on ecological and sustainable principles is urgently needed.

In areas outside Moldsilva's formal influence, such as publicly owned lands of administrative-territorial units (ATUs) or the 13.6% of total forestland and privately owned forests or 0.7% of total forestland [42], forest management remains rather inadequate for supporting sustainable ecosystem development as most of them do not even have forest management plans (FMPs) that are mandatory or such plans are outdated [43]. Moldova's forests are still facing the problem of illegal logging, and its main causes are driven by a high level of poverty associated with a high and continuously increasing price of firewood, from 340 MDL/m$^3$ (18.6 USD) in 2010 to 530 MDL/m$^3$ (26.7 USD) in 2016, which is hardly affordable for the rural population [32]. Moldova faces an unsatisfied demand for firewood and inefficient or even a lack of guarding efforts, and randomized control

activities undertaken by forest or environmental institutions still reveal impressive volumes of uncountable wood.

According to an FLEG analysis, the forest ecosystems most affected by illegalities were those managed by the ATUs [44]. Although for the period 2009–2014 nearly 5 million MDL (0.3 million USD) was recovered through fines for the damage caused by illegal logging, the annual damage to the state budget was about 45.5 million MDL (2.9 million USD) [45], while the damage to ES was estimated at much higher figures reaching about 8.8 million USD [46]. Another FLEG analytical study conducted in 2010/2011 in cooperation with Moldsilva estimated an annual wood consumption in households at 1273.7 thousand m$^3$ [47], of which the firewood and associated agricultural biomass constituted the most of it—1036.5 thousand m$^3$. At the same time, the National Bureau of Statistics (NBS) provides another figure for firewood consumption for 2015–2016—2405.7 thousand m$^3$, with an average consumption of 3.6 m$^3$ per households in rural areas [48]. Such differences in data primarily caused by the different methodologies applied make this social need for firewood (or energetic wood) difficult to assess.

In terms of socio-economic development and living standards in rural Moldova, a positive aspect is the downward trend in the absolute poverty rate, from 39.5% in 2014 to 35.3% in 2020 [49]. The total average monthly disposable income per person increased from 1477.2 MDL (94.1 USD) in 2014 (Figure 1), which did not cover the subsistence minimum, to 2702.3 MDL (156.7 USD) in 2020, which exceeds this minimum by 699.5 MDL (40.6 USD; NBS, 2021). According to the same NBS sources, the subsistence minimum is the minimum volume of products and services necessary to meet basic needs that ensures the maintenance of health and sustains human vitality.

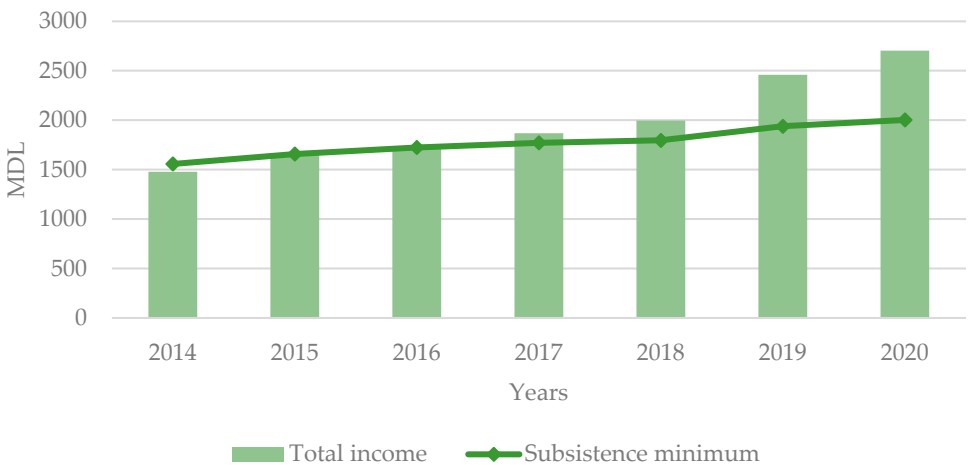

**Figure 1.** The monthly average value of the total income and the subsistence minimum per person in the rural area of the Republic of Moldova [49].

Of the total disposable income of the rural population, wages have the largest share, reaching 40.5% in 2020, compared to only 30.6% in 2014 (Figure 2) [49]. The increase in the contribution of wages to total income has also been influenced by the increase in the average gross monthly wage. For the three districts of the selected communities under this study, the wage increased from 3353.0 MDL (213.6 USD) in 2014 to 6281.6 MDL (364.3 USD) in 2020. The share of remittances in total income decreased from 24.8% in 2014 to 16.1% in 2020, while the share of social benefits (such as pensions, child allowances, social aid, etc.) in total income increased from 17% in 2014 to 20.7% in 2020 [49]. The size of the average retirement account for old-age people has been continuously increasing and amounted for the districts of sampled communities to 943.97 MDL (60.1 USD) in 2014, reaching 1591.02 MDL (92.3 USD) in 2020 [49].

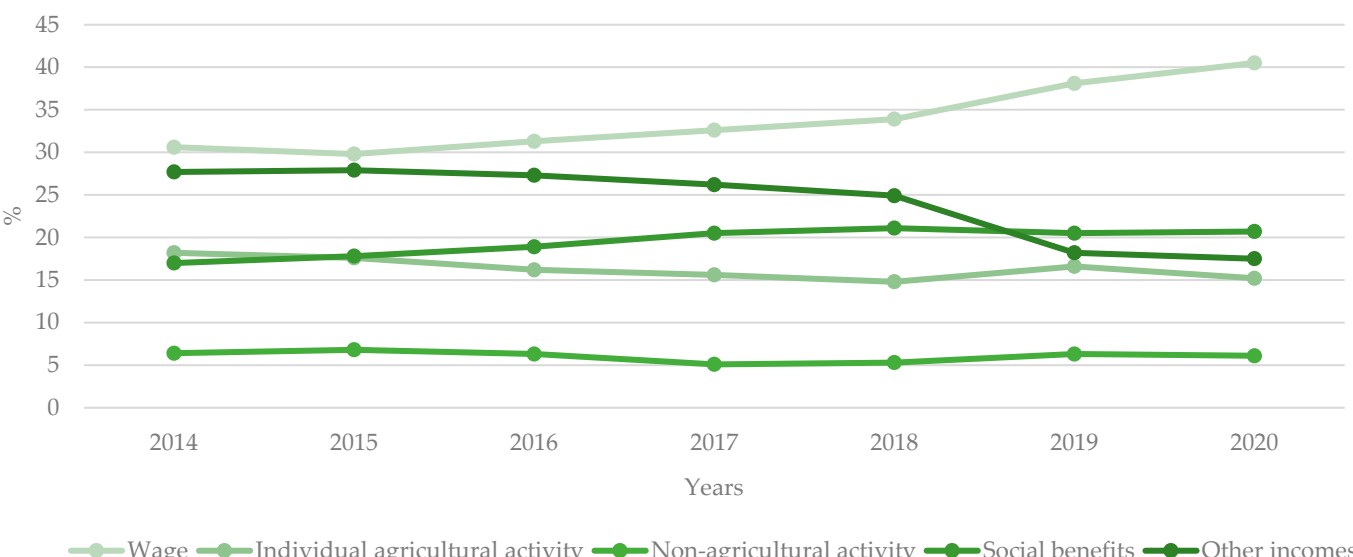

**Figure 2.** Sources of income of the rural population in the Republic of Moldova [49].

Non-agricultural activity [49] made an approximately equal contribution to total income during the reference period (2014–2020), while individual agricultural activity (mainly crop cultivation and livestock sector) declined, accounting for only 15.2% of total income of the rural population compared to 18.2% in 2014.

For Moldova, the term 'household' is of a sheer historic and cultural significance. With most of its population living in rural areas, Moldova's countryside is also a generator of various livelihood products for the whole population. In addition, many households are in urban areas (within so called municipalities' property), but they are not purely cities, so they share almost similar socio-economic (and forest dependency) features as households outside of a city's influence. The number of households registered in 2014 according to the latest population and housing census was 959 thousand [50], with an average size of 2.9 persons.

## 3. Study Area and Data

To conduct comparative assessments, in this study, we considered the same three communities subject to the 2014 survey [27] that were selected to best reflect the acuteness and sensitivity of forest dependency, also including the traditional occupations of the local people (Figure 3):

1.  Alexandru cel Bun village, which is part of the Volovița commune (a collection of two villages) in the Soroca district, also part of the Nistru river basin and of an Emerald site in the Northern region (a forest-steppe landscape type);
2.  Ciorești village, which is part of the Ciorești commune (a collection of two villages) in the Nisporeni district (central region or central hilly plateau, part of the most forested area in the country);
3.  Borceag village, a locality in the Cahul district of the Southern region (part of a steppe landscape with scattered forest vegetation).

The selection of these localities was based on eco-geographical representativeness [27] and according to the three distinct regions of the country—North, Centre, and South, where forest distribution per each region is 26%, 58%, and 16% respectively [42]. The proximity to forests as a criterion affecting accessibility for an individual or community to be forest dependent was taken into consideration to better understand the relationships on an occupied (inhabited) versus unoccupied forest area [27]. More specifically, Alexandru cel Bun village is located close to riverbank forests composed of pedunculate oak; Ciorești village is in the heart of ecosystems with old-growth forests of pedunculate and sessile oaks; and Borceag village is part of a silvopastoral landscape with pedunculate and downy oaks

as main species in the scattered forests. All the mentioned oak-type stands/habitats are extremely valuable forest ecosystems [35], also given the recent trend in the country's forest management where areas covered with native oaks almost match the areas covered with introduced/exotic (but close to naturalized) acacia species dominated by black locust [36].

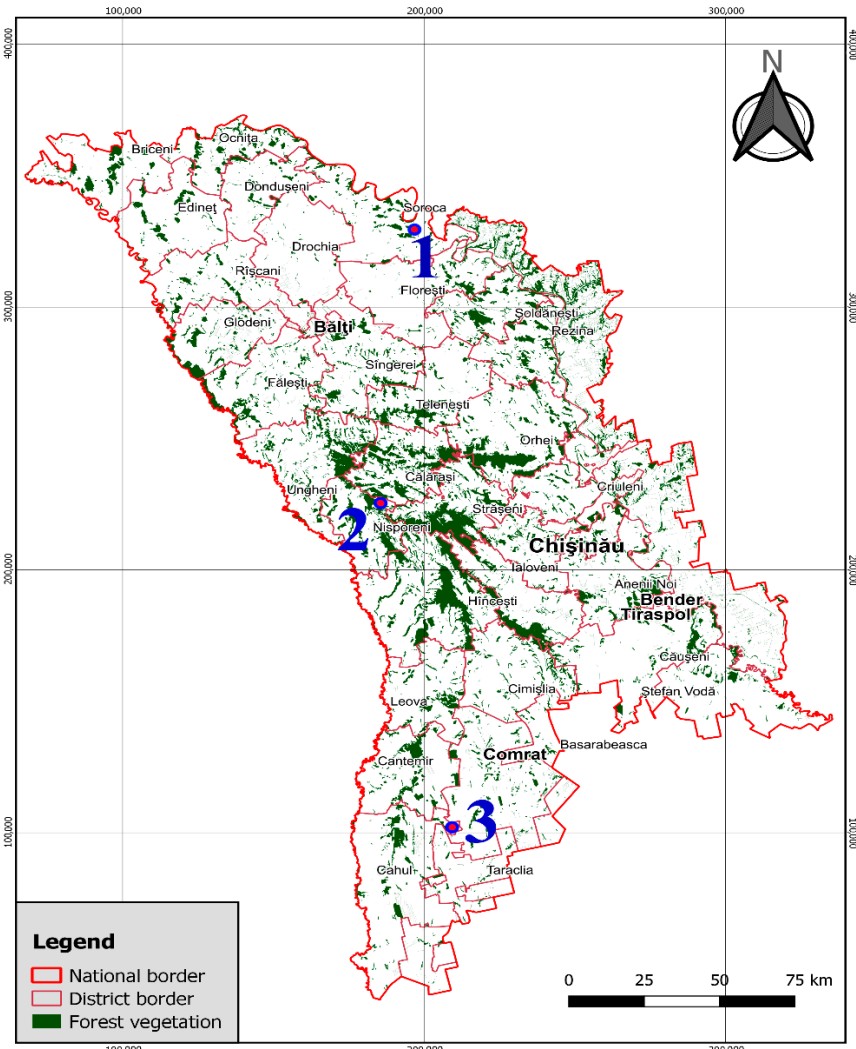

**Figure 3.** Eco-geographical position of the three selected localities considered for this study (1—Alexandru cel Bun, 2—Ciorești, 3—Borceag).

With regard to land use, the main occupation in all three villages is agriculture. The majority of land is in use by so-called peasant households. These are individual enterprises, based on private ownership of the agricultural land and other assets, supported by the personal labor of family members (who are members of the peasant household), that aim at obtaining agricultural products and handle their primary processing and market mainly their own agricultural products [51]. Concerning animal husbandry and grazing, many rural households engage in collective privately-led farming on public pastures (often grazing in forests, which is illegal), but the poor can be limited in accessing forage.

The village of Alexandru cel Bun, part of the Volovița commune, covers an area of 0.86 km² with a perimeter of 5.05 km, located 7 km away from Soroca town (a district capital city) in northeastern Moldova. According to official statistics [42], agricultural land has the largest share (57.7%) in the commune territory; only 1% of this land is in active use by private individuals (households with their own gardens). The rest of the area is processed by peasant households and a limited liability company (LLC), which process their own land and land leased from individuals. Forest land occupies only 5.2% (79.23 ha)

of the commune's territory. There are two important forest bodies in the proximity of the village: 657.45 ha and 557.73 ha, both owned and managed by the state forest enterprise Soroca under Moldsilva.

The village of Ciorești, part of the cognominal commune, covers an area of 2.99 km$^2$ with a perimeter of 12.05 km, located 20 km away from Nisporeni town (a district capital city) in the central hilly (forest) plateau of the country. The commune has a considerable area of forest land (38.7%) surrounding it, while the share of agricultural land is 46.5% [42]. Nearly 20% of agricultural land is used for work by individuals in their own gardens, and there is almost no private land offered for rent. The remaining agricultural land is in use by peasant households and by an LLC. The commune leaves an impression that it is surrounded by hilly forests, and from an ecological point of view, these forests are part of the typical natural forest ecosystems of central Moldova (a historical region called "Codri", meaning "forests").

Borceag village is a community in Cahul district (southern Moldova) covering an area of 2.28 km$^2$ with a perimeter of 12.48 km, located 28 km away from Cahul town (district city). Compared to the other two localities included in the study, it has the highest percentage of land used for agriculture (60.5%), of which only 4.9% comprises gardens managed by individuals [42]. Nearly 75.6 % of agricultural land is leased to several LLCs, while the rest of the 24.4% is managed by 184 peasant households that own their agricultural land. The village has its own communal forests (circa 45 ha, on which 35 ha are sporadic forest plantations and 10 ha are shelterbelts). All community-owned forest vegetation is mainly composed of shrubby and tree species, mostly planted acacias. Nearly 14% of the land is state owned forest land managed by the state forest enterprise "Silva-Sud" under Moldsilva [42].

Moldova's forest sector is dominated by state public ownership (81%) which is managed by Moldsilva and its network of forest entities. Members of all three sampled localities have access to Moldsilva's forestlands, and they either participate in various forest-related activities (such as planting or harvesting) or gain other benefits (such as berry/fruit/herb collection, recreation, hunting, etc.). Moldsilva's lands have mandatory FMPs based on sustainable principles, and these documents are updated on a 10-year period basis [35]. Forest vegetation outside Moldsilva, mainly owned and managed by local communities, is in worse condition and existing FMPs are insufficiently enforced or not enforced at all (though Moldsilva's entities are cooperating with communities on forest management planning).

## 4. Materials and Methods

People's dependence on forests is directly linked to household subsistence needs, primarily livelihood-oriented strategies such as providing firewood, timber, NTFPs, jobs, and ES [22]. In Moldova, with limited forest areas but of great traditional significance, all forest ecosystems provide a range of benefits to local communities, and their values significantly exceed official figures of what the actual provisions forests display [52]. To assess this dependence on the forest in the three selected localities, two types of questionnaires were developed und used: (i) a standard quantitative analysis questionnaire (questionnaire 1) for assessing the income of each individual household surveyed (Table 1), and (ii) a technical questionnaire (questionnaire 2) for more rather qualitative data (Table 2).

A similar research approach has been addressed in several other studies undertaken in Eastern Europe, Southern Caucasus, and Russia. The questionnaires we used were jointly adapted for these regions based on specific elements provided by the World Bank Living Standards Measurement Survey and CIFOR PEN [25]. The main purpose of this survey was to assess the contribution of forest and environmental resources to the total income of rural households. According to the PEN guidelines [53], a total income is defined as the sum of forest and non-forest income minus the costs of purchased inputs, which also emphasizes that households' subsistence extraction and production (meaning in addition to extraction or production that generates cash income) should be included in total income too.

**Table 1.** Sections and content of the household questionnaire 1 (based on [54]).

| Sections | Title | Content |
|---|---|---|
| 1A | Basic information on household members | Relationship to head of household, gender, age in years, years of education, main and secondary occupation of members $\geq$ 16 years |
| 1B | Identification of the main respondent | Which household member was interviewed |
| 2A | Fixed assets (land) | Area of land controlled or used by the household |
| 2B | Other household fixed assets | Other fixed assets of the household, their quantity and age |
| 3 | Forest resource base and environmental services | Distance from the forest, planting of forest trees on own farmland and purpose, perception of ES (respondents were asked to give a score from 1 to 3) |
| 4A | Forest and environmental income | Quantities of forest and environmental products |
| 4B | Firwood consumption | Quantity of firewood consumed |
| 5A | Income from agriculture | Quantities of agricultural products |
| 5B | Costs in farming | Costs of agricultural production |
| 6A | Livestock and their income | Keeping, consumption, and sale of livestock |
| 6B | Income from animal products | Quantities of products of animal origin |
| 6C | Costs in animal husbandry | Quantities and value of inputs used in livestock production |
| 7 | Income from salaries | Total income from wages and salaries for each household member, including seasonal work |
| 8 | Business income | Total income from own business |
| 9 | Other income | Amount received during the last year for each source of income |

**Table 2.** Sections and content of the additional questionnaire 2 for representative persons (based on [54]).

| Sections | Title | Content |
|---|---|---|
| 1 | Most important product | For each product category, respondents were asked: the most important products for the livelihood of the rural community, changes in availability and their causes, suggestions for actions to increase their availability (respondents were asked to give a score from 1 to 3) |
| 2 | Seasonal calendar | The months in which the most important forest or environmental products are harvested, and which are the most important seasons for agricultural activities |
| 3 | Infrastructure and markets | Number of roads, access to electricity, gas and water, distance of villages from markets, other benefits received related to forest services |
| 4 | Wages | Regular wages for men/women in good times/hard times |
| 5 | Prices | Local prices for products in the village |

An interview method using questionnaire 1 was applied—one questionnaire per one household (one to one system). Interviews lasted anywhere from 30 to 40 min, but usually

not more than one hour. The period of interviewing through questionnaires took place in August and September of 2020.

Households where interviews occurred were selected using the three-household sampling design. After interviewing the respondent from the first household located at the entrance to the locality, the third household in order was to be approached. If members of the household refused to respond, the one in the immediate vicinity was approached. Members of 50 households were interviewed in each village, resulting in a total of 150 interviews. Besides the standard questionnaire, for 10 representative persons in each village who occupy higher social or technical positions, questionnaire 2 was used (Table 2) to collect qualitative data on the most important livelihood products as well as more information on infrastructure, markets, and prices of products. For this, three persons who more adequately knew the general situation in each village were approached: town hall employee, social worker, and forester. After interviewing these persons, they were asked to give contacts of other people who possessed adequate knowledge pertaining to the situation in that village.

During data collection of the previous 2014 interview analysis [27], the authors encountered a problem, the so-called 'survey refusal', whereby respondents were rather reluctant to answer section 2B of questionnaire 1, i.e., questions regarding the goods owned by household members and the value of these products. In this study, the monetary part was avoided, so only questions regarding the presence of the main groups of goods (such as car, tractor, motorcycle, TV, refrigerator, washing machine, oven, computer, stove, bed, cupboard, table, etc.) along with their quantity and age were applied. The current value of the goods was then estimated based on a grid which contained indicative prices for each type of product for 1, 5, 10, 15, and 20 years of age. Thus, most members of the households interviewed responded to this section.

All data collected from offered questionnaires were then entered into an MS Office EXCEL database. With the help of questionnaire 2, based on the answers of the total of 30 respondents (i.e., information concerning the prices of agricultural, livestock, and forestry products present in each village), a table with average reference prices for each type of products was produced. Using this table, data on quantities of products in all categories were converted into monetary value, and by subtracting the costs for their production (reported by the respondents), a value of income per source of income was generated. To identify the average monthly income per person, the total income earned per village was divided by the number of adults in each village, excluding those still in education and not contributing any income to their households. To assess the influence of income on forest dependency, households were sorted by income level, with the series divided into five equal parts. The first quintile comprises households in which members have the lowest income, and the fifth those with the highest income.

The 2014 research [27] included both income and expenditure incurred by procuring forest products in only one category, and that is the income for household members. In the present study, the dependence of households on the forest was analyzed by considering both collected and procured products. On the one hand, some villagers receive forest products as social aid (either for free or against a symbolic reward) or collect them directly from the forest (meaning they are allowed to have free access to the resource), which they value themselves or consume in their own households—these are *collected products*. On the other hand, firewood as a primary energy source is the main demanded forest product villagers procure against a fee proscribed in the technical norms or other economic documentation of the state forest entities subordinated to Moldsilva—these are called *procured products*.

## 5. Results

### 5.1. The Role of Forest Ecosystem Services as Seen by Local People

From the total 150 questionnaires completed, two were excluded from the study due to incomplete information provided by the respondents—one per each of the Ciorești and

Borceag villages. The rest of the 148 questionnaires allowed us to continue processing data and assess the income level of the studied households.

In questionnaire 1, respondents were asked whether household members had planted forest trees on their own farmland in the last 5 years, and from a list containing several possible purposes, they chose the three most important types of plantings. In addition to these questions, to identify people's perceptions of ES (provisioning, regulating, and cultural), questions were asked about what household members consider a forest ecosystem provides for them, how it helps, and why the forest is important to them. Of all the respondents, only one person said that members of their household planted forest trees on their own farmland. The main purpose of the planting was firewood for their own use, which would increase the value of the land and allow children and grandchildren to own these trees.

In terms of provisioning ES supply, members of the households surveyed consider that the forest ecosystem primarily provides them with food (2.8 out of 3; they collect mushrooms and berries), fresh water (2.7 out of 3; springs are the main source of drinking water for their villages), and natural medicines (2.7; such as gathering medicinal plants, usually meaning tea making or ingredients added to a meal). Less importance was given to fuel—although they are dependent on wood resources (2.4 out of 3) to heat their homes, people do not consider the ecosystem to provide them directly with firewood as they must buy it from state forest entities. Local communities do not see the forest as a provider of genetic and ornamental resources; thus, all the respondents scored these two with 1 out of 3.

Regulatory ES are highly appreciated by the surveyed population. It is generally perceived that forests can help regulate/control water quality, stabilize climate and water runoff, reduce natural disasters, combat erosion, enhance soil formation, improve natural water purification, and reduce costs for water treatment. Cultural services are also considered important as forests allow access for recreation (2.9 out of 3), bearing a high aesthetic value (2.8 out of 3) and being important for cultural heritage too (2.6 out of 3).

Questionnaire 2 helped us identify the most important products for the well-being of the surveyed villagers. The respondents chose the main products from the following categories: wood products, forest food, and other forest products. For the chosen products, opinions were received on the availability of these resources, whether it has increased/stayed or the same/decreased, and the magnitude of the factors influencing it, rated on a scale of one to three. Of the wood products, firewood is considered the most important. Although this resource brings virtually no income, but rather expenses, it is seen by local people as an important factor conditioning their livelihood (all respondents were unanimous on this point). Generally, wood availability is perceived to have been decreasing over the last five years. Although people acquire the necessary quantity from state forest units every year, they have a negative opinion toward forest managers, considering that forest exploitation is carried out without control. For example, in the village of Ciorești, 8 out of 10 respondents indicated that the main reason for the reduced availability of firewood is over-exploitation. This perception was countered by another respondent, a retired former forester, who said that the amount of wood available for harvesting has remained within the same limits from year to year and that all the harvested wood is meant to supply the social demand for firewood.

Important products are also berries (as food from the forest) which, according to respondents, have decreased in availability, the main cause being drought in recent years (2.8 out of 3), then increased use due to over-harvesting by villagers (2.4 out of 3), and uncontrolled/unsustainable primitive harvesting practices resulting in damage to plants/branches) (2.1 out of 3). Other important products are medicinal plants and water, so their availability was described as being maintained at the same level—eventually, there were no responses from the respondents that their availability had increased.

From the perspective of the locals, the most important factor to increase the use or the income from the most demanded products was the facilitation of access to the forest and the

provision of more rights to the users (2.8 out of 3). Another factor was better protection of the forest by avoiding overuse (2.4 out of 3), due to the common negative perception of the population toward forest managers. No one considered that better access to capital/credit and equipment/technology for harvesting and processing would condition increased use or income.

As for ecotourism development, after consultation with the mayor of the village of Ciorești we discovered that the commune has implemented a project entitled "Prosperous tourism means developed localities", which helped launch a touristic route coined "Hanker forest" (in Romanian "Dor de codru").

*5.2. Sources of Income*

Our analysis shows that the largest contribution to total income of the villages' surveyed population is made by wages, including remuneration for unskilled labor (Figure 4), with the highest percentage for the village of Ciorești.

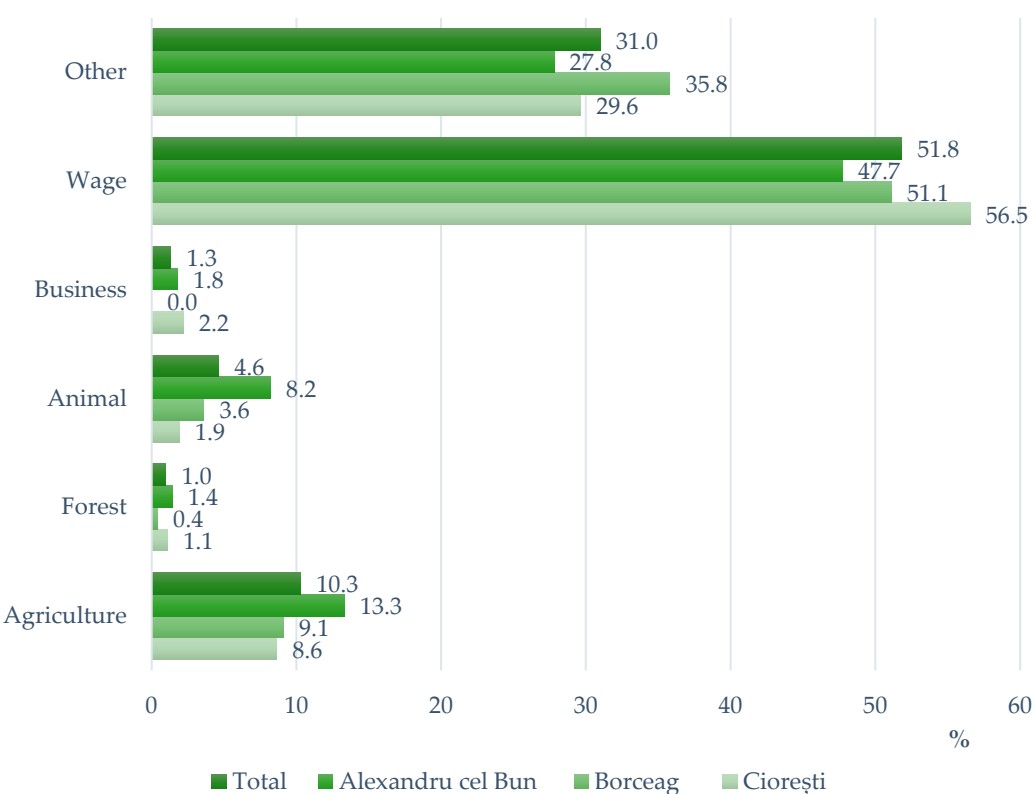

**Figure 4.** Annual sources of income and their share of total income.

The resulting average monthly income per person for the three villages is 2710.7 MDL (157.2 USD), which is comparable with the value of the same indicator for the rural population provided by the NBS for 2020. For the village of Alexandru cel Bun, the resulting average monthly income per household is 2825.7 MDL (163.9 USD) and for the other two villages this indicator is smaller, for Borceag at 2722.0 MDL (157.9 USD) and for Ciorești—2584.5 MDL (149.9 USD).

Income from other sources is the category that has a significant share: 31% of the total income of all interviewed households. In the other income category, retirements account for 48.2% (14.8% of total income) and remittances for 38% (11.7% of total income). Material aid received from the government (child benefit, disability, and social assistance) contributes to 8.3%. In the village of Borceag, respondents indicated that they also received 3 m$^3$ of firewood as social aid every winter (11 households responded). Household members offer their own agricultural land for rent, and as a reward they receive either cash or cereals (or similar products), except in the village of Ciorești where such practices were not identified.

### 5.3. Forest Products and Their Collection

Timber products are generally only accessible to people based on a commercial relationship with the forest managers. Only three cases were reported in the questionnaires where household members collected insignificant quantities of twigs/branches without paying for them, although most responded that this practice was not legal. In two other cases, branches were offered as remuneration for seasonal work, and in the remaining cases, branches were purchased. Most of the wood is purchased from state forest units or received as social aid (in case of Borceag village for firewood).

The NTFPs are generally available to locals (compared with restricted accessibility to timber resources), but only some (such as mushrooms, forest tree seeds, berries, and partly medicinal plants) are actually collected from the forests or trees. Walnuts, (*Juglans regia*), which have the highest total value (Table 3), are collected either from trees growing within households or shelterbelts along roads. Almost 90% of the nuts (mainly walnuts) are collected for sale, the rest are used as food for the locals' own consumption. Dogrose fruits and other medicinal plants are mainly used for food (tea making and ingredients), and are mostly harvested from agricultural land or from spontaneous flora around forest edges, so all these have a low monetary value.

**Table 3.** Frequency and total value of forest products collected and procured.

| Category | Frequency | | Total Value | |
|---|---|---|---|---|
| | **Number** | **%** | **MDL (USD)** | **%** |
| Firewood, of which | 140 | 44.59 | 779,200 (45,193.6) | 87.55 |
| Collected | 11 | - | 35,200 (2041.6) | - |
| Procured | 129 | - | 744,000 (43,152.0) | - |
| Branches, of which | 8 | 2.55 | 14,070 (816.1) | 1.58 |
| Collected | 5 | - | 4950 (287.1) | - |
| Procured | 3 | - | 9100 (527.8) | - |
| Nuts | 37 | 11.78 | 79,375 (4603.8) | 8.92 |
| Mushrooms | 41 | 13.06 | 10,340 (599.7) | 1.16 |
| Medicinal herbs | 40 | 12.74 | 2475 (143.5) | 0.28 |
| Dogrose | 35 | 11.15 | 1754 (101.7) | 0.20 |
| Forest fruits | 9 | 2.87 | 1405 (81.5) | 0.16 |
| Seeds | 4 | 1.27 | 1420 (82.4) | 0.16 |
| TOTAL | 314 | 100 | 890,039 (51,622.3) | 100 |

On the other hand, locals can directly cooperate with Moldsilva's units by selling them certain quantities of home-grown tree seeds, medicinal plants, strawberries, etc. Moldsilva units welcome this relationship as they have the duty of harvesting NTFPs (fruit and berries, medicinal plants, agricultural and animal products, bee honey, vine snails, fish, etc.) which they must fulfil according to set plans, so they gladly buy various NTFPs from locals or directly employ local people who are paid for harvesting these products from their own forests or from other available lands.

Wood is a heavy burden on rural households and a large share of local people's income goes to purchasing their required wood resources (Figure 5). The most disadvantaged are low-income households, with members spending 18.8% of their income on wood. Wealthier households are less affected, with only 3.9% of their income allocated to the purchase of firewood, although wood consumption is roughly the same for all social categories.

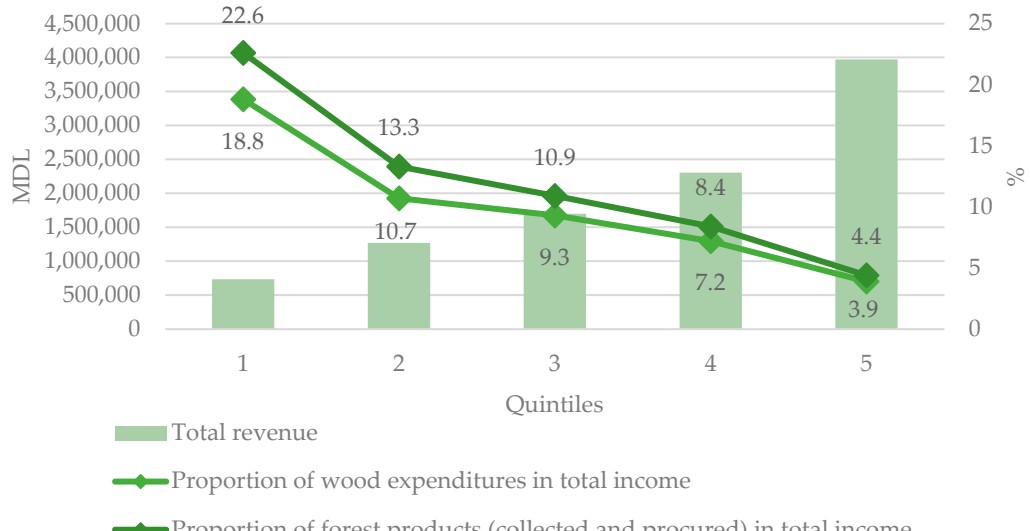

**Figure 5.** Dependence on forest (collected plus procured products) and proportion of expenditure on procured products (wood) in total income by income quintile (expenses on wood and forest products value are associated with percent scale on the right).

The annual wood consumption in the surveyed households was estimated at 731 $m^3$ with an average consumption per household of 5.2 $m^3$, which is approximately the same amount as in the 2014 report's value of 718 $m^3$. For Alexandru cel Bun village, the average wood consumption per household was 4.8 $m^3$; 20 respondents said that they used coal (12.4 tons in total) in addition to firewood, two other households also used pellets, one household used only natural gas for heating/cooking, while two respondents indicated they used 4 and 2 $m^3$ for making barrels (for wine) or similar housing needs (though those amounts of wood were purchased as firewood). For the Ciorești village with the highest average of annual wood consumption of 6.2 $m^3$ per household, two households used only branches to heat their houses, and only one household used exclusively natural gas. In the Borceag village, the average of annual wood consumption was the lowest and was estimated at 4.7 $m^3$—the majority of households used firewood, one household used biochar in addition to firewood, two used only natural gas, and one used only biomass briquettes.

As income increases, changes in the use of forest products follow (Figure 6), with wealthier households using more products. The share of forest products actually harvested and used is higher in middle-income households. However, the lower the household income is, the higher the dependence on the forest turns out to be, which is explained by the fact that forest resources constitute a significant share of the total income in the surveyed households (Figure 5).

Analyzing age distribution among household members per each of the five income quintiles, the average age of the household heads is the highest in the first quintile with most (70%) of them over 60 years old, but it decreases toward the fifth quintile with an average age of 52.7 years old.

The respondents avoided answers about money, but most of them were willing to answer the questionnaire's section about the fixed assets their households owned (except for five respondents who refused to provide such information). The amounts of fixed assets owned per household were divided into quintiles and based on the total household wealth and the value of forest resources, the dependence on forest resources was assessed according to household wealth. The results showed the value of forest resources used in households with more modest wealth represents a higher share of total income (Figure 7), compared to more affluent households where this share is lower.

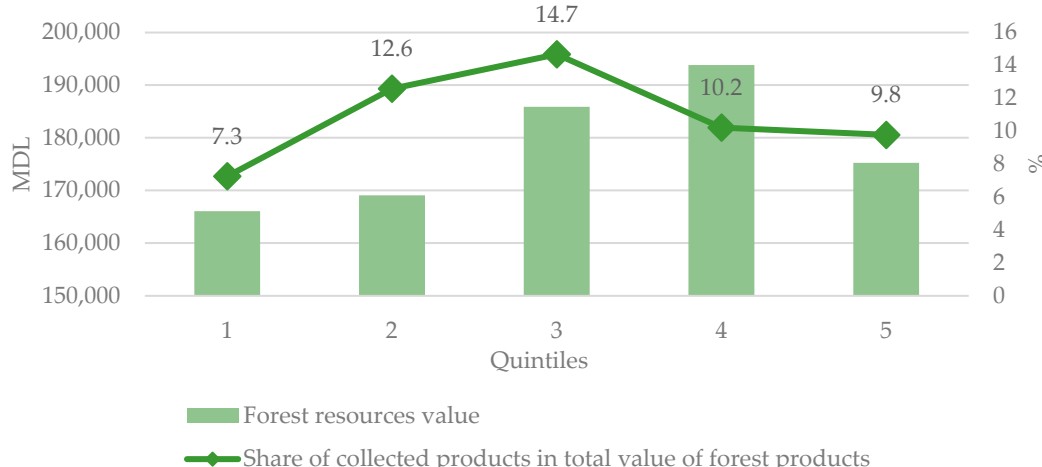

**Figure 6.** Share of collected products in total value of forest products (collected plus procured) by income quintile.

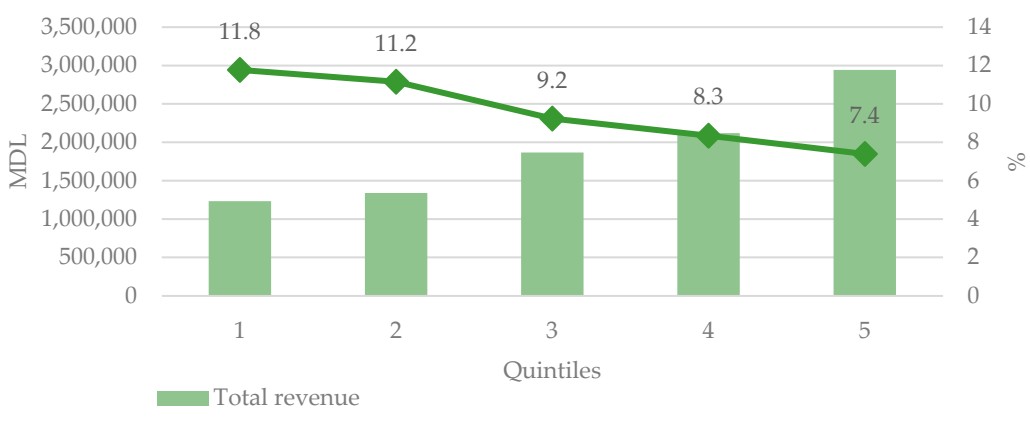

**Figure 7.** Forest dependence (products collected and products purchased) of households in sampled villages by wealth quintile.

Regarding years of education, the average for the first quintile was 9.8 years with the highest score of the household head (12.3 years) in the fifth quintile, those with a higher income level. The average number of years of education was calculated for members of each household who were no longer in education, these were placed in one of three categories: (1) less than and including 9 years of education (22% of households), (2) more than 9 years but less than and including 12 years (37% of households), and (3) more than 12 years (41% of households). As the number of years of education increased, the share of the value of forest products used in total income decreased. In the first category the share was 11.4%, and in the third category it was 7.4%.

In terms of distance from the forest, households located at distances of less than 2 km have approximately the same share of forest products in total income (9.8% and 9.3%), and they are more dependent on these resources than households located between 2 and 3 km from the nearest forest (6.5% of total income).

## 6. Discussion

Forests provide an essential contribution to the incomes of both rural and urban communities, and in many cases this contribution is not recorded in national statistics [4]. Moldova ranks among the poorest countries in Europe [55] where rural populations prevail (58%) [56]. Authorities struggle to boost economic development (and even EU accession is

on the political agenda), but there is currently no clear understanding of how forests may help the poor reduce poverty or improve their economic conditions [36]. Whatever the scale of forest management issues, Moldova's scattered forest lands/bodies have always been, without a doubt, the main salvation for the rural population for centuries, especially in years of energy or economic crisis. The lack of trustworthy information about Moldovan forests' contribution to human wellbeing implies the need for a tool that could identify the extent to which forests contribute to improving the living standards of its people, especially in rural areas. To this end, the use of quantitative analysis questionnaires in this assessment provides a potential source of data for designing policies in the forestry sector, including inter-sectorial vision where social and economic (primarily energetic) values dominate. The difficulty in applying these types of questionnaires to rural dwellers is that the population is still reluctant to answer what income they actually have, which raises certain questions by opening doors to other types of research. Despite this reluctance to express themselves in monetary values, people were observed to be freer to say what types of products they produce/harvest and in what quantities. The monetary value of these products was then quantified based on the average price (identified using the questionnaire 2) and the declared quantities. Although this conversion may be subject to error, the share of different sources of income in total household income is comparable to that provided by the official statistics [49]. However, the difference is observed in the share of wages in total income, and in this study, this is explained by the fact that the bulk of the values coming from non-agricultural activity was included in the wage activity. Thus, we can state that the data are at least close to the real situation and can be used to describe the household–forest relationship.

The results of the survey show that the largest share of rural households' income comes from wages, 22% higher than in the 2014 survey. Wage income is the main income in rural areas elsewhere in the region, according to analyses in the former Soviet countries of the South Caucasus (Armenia, Azerbaijan, and Georgia), Eastern Europe (Belarus and Ukraine), and Russia [25]. The income from forest resources is rather low, but forest dependency is more than just direct income from forests, and it seems to be important for both the poorer population (which is, however, more dependent on forest resource) and the richer ones, though for the latter it is an additional income [57–59]. This income is described as an important one for the countries in the region, even though forest resources contribute with only 4–8% to the total income, and 16% in Georgia alone [25].

## 6.1. Ecosystem Services from a Rural Perspective

In addition to traditional questions about the income that local household members receive from forest resources [25], in our study, we used questions about people's perception of ES and whether they feel they can actually benefit from these forest services. This seems to be crucial for both ordinary people and decision-making bodies (at both local and central governments) as the continuation of the current unsustainable forest management along with an ignored critical investment in ES will likely cause long-term economic loses [60].

All the values associated with ES result from people's daily interactions with the environment they live in, so people perceive the environment primarily based on the socio-cultural setting [20]. In our study, we assumed that the general appreciation and enjoyment of ES provided by forests among the members of rural households was rather high, at least at the level of the traditional human–nature connection. People's reliance on forest provisioning services for their livelihoods is rather well captured, especially in lower wealth groups [61]. In our study, people directly indicated that other types of services are important too. For most Moldovans, picking a specific type of ES is directly influenced by their access to the main forest resource of their livelihood (which is firewood) and indirectly by the perception of a declined availability of non-timber resources. We noticed a general understanding among respondents who seemed to agree on the forests' key role to protect them from natural disasters. Cultural services are also deeply rooted in the traditions and culture of rural communities. Local populations recognize the aesthetic value that forests

bring to the landscape, and that forests also play an irreplaceable role in the individuality of their own settlements. For communities, a forest in their immediate vicinity means a place of recreation, tranquility, inspiration, and education (children are traditionally taken on excursions to discover the beauty and diversity of nature).

*6.2. Dependence on Non-Timber Forest Resources*

The people of Moldova have free access to NTFPs (unless it is for commercial use), but in our study, income from forest resources represented only 1% of the total income of rural households (Figure 4), compared to 11.3% in 2014 (while only income from non-timber resources was 6.6%). However, the 'free access' to forests does not mean that adequate conditions for recreation or other activities are provided [30]. Recreational activity in forests is not well organized and is often controversial as the former practice of forest leasing for recreation and hunting was just another 'legalized' method of forest fragmentation [28]. The free access to forests competes with climate change, especially through the more frequent droughts in recent years, and the population receives little benefit from a slightly changed forest environment because of the dry seasons. Under conditions where the area (and quality) of forest crops is shrinking, local populations must also compete with Moldsilva's units who have their own plans to collect NTFPs (mainly berries and medicinal plants or whatever they can potentially collect from forests).

The local population is not encouraged, nor are the state forest entities motivated, to promote the production of NTFPs. Moldova is an example of how the decline in NTPF availability (including game) can hamper the forestry sector's proper development. A continuous decrease of interest in NTFPs is the result of a combination of factors, such as the poor maintenance of the existing production areas, the lack of investment in creating new plantations, the absence of innovative technology for harvesting and processing, and high taxes for collection and logistics [39]. As Moldsilva is a self-managed [33] and a self-financing organization, most of its expenses (nearly 98% of the annual turnover) are covered by the income from the sale of products and the provision of various services (mainly wood/timber sale). Moldsilva, obtaining revenues almost entirely from wood products as the main provider of timber for the domestic market [35], is no longer motivated to generate more income from other forest activities, although salaries/wages in the forest sector remain below the average per national economy and the number of employees is continuously decreasing [35]. The forest sector is still offering permanent and temporary jobs for nearly 4000–5000 people [30]. Such a monopolistic position of Moldsilva's entities discourages sectoral competition and does not create incentives to find alternative solutions [34]. Another issue is the availability of NTFPs, considered to be decreasing due to increased resident collection (a problem also identified in 2014, which is still valid today). Almost all NTFPs are still collected from the forest without studies on the biological/ecological and commercial capacity [39].

People who participate in the collection of NTFPs are affected by low market prices for their unprocessed products, suggesting that new processing facilities would bring additional income and create more jobs [62]. In the absence of proper assessments for NTFP sustainable production capacity, it is difficult to forecast how much local people can be sustained through an additional income from non-wood products. Assuming that the effects of harvesting NTFPs are less destructive than harvesting wood products [24,63], there may be alternatives to optimize the situation through the establishment of new (forest fruit) plantations and the creation of more opportunities for the local people (e.g., Moldsilva could procure higher amounts of non-wood products directly from the local population that is allowed to harvest their own lands). New and better markets can bring higher prices [62], and the increased availability of NTFPs can also be influenced, as in case of Azerbaijan [58], where new equipment and processing technologies increased the availability of hazelnuts in forests. Managing non-timber initiatives is challenging, but NTFPs can be an essential support for forest-dependent rural households, especially in times of crisis [64]. Importantly, for those who plan such projects, the use of both timber

and non-timber forest products can be compatible if the forests or plantations of high-value species are not overexploited.

*6.3. Dependence on Timber Forest Resources*

For Moldova, firewood is a major social necessity, and thus people must pay considerable amounts for it compared to their factual income. The most affected the are low-income households. The richer the family, the lower the share of expenditure in their total income. Firewood has been considered the most important wood product in all studies in the former Soviet countries [25], and access to this resource has been restricted in most cases. Even in forest-rich countries in the neighborhood, such as the Belarus [62], locals are concerned about any potential decrease in the availability of wood resources, and they believe that providing freer access to the forest will increase the level of wood use. The same is true in Moldova, where people think the access to public forests is limited, but if they were allowed greater access to forests, they would collect more wood. However, without free access to forests the people dependent on wood will purchase it from more suspicious sources or even from illegal practices [58,59,65]. As gas prices continuously rise [59] and the unaffordability of gas increases correspondingly [66], firewood remains the most important source of heating (and cooking) for the rural population in Moldova's households. We found that even if people claimed they have access to a gas pipeline as an alternative to heating their homes, they had to turn to wood heating because of the overly high price for gas. A higher gas price means fewer people who can afford it, which in turn, means more pressure on existing forest ecosystems through various schemes, including unsustainable or illegal harvesting. Moldovan forests are considered to be managed irrationally [67] and their overexploitation is the result of a complex of factors difficult to address [36], which calls for a cross-sectoral approach. Throughout the course of our comparative investigation, it was not possible to formally identify whether local people buy only legal wood. We received many informal hints on possible illegalities, which matches data from a FLEG psycho-sociological analysis [68] where most respondents did not accept illegal wood trading with Moldsilva's units, but they admitted buying wood at lower prices even though they knew the wood was illegally sourced. In our study, all the respondents claimed that wood can only be procured officially from Moldsilva's forest entities.

Most of the surveyed households had outdated wood burning stoves or heating systems. Wood is usually procured in late autumn when it is still green (or wet), which is inefficient because much of its energy is wasted via the evaporation of its moisture. The need for more efficient heating systems that reduce fuel bills (and emissions) and save consumers' money is another awareness challenge local people face. The highest wood consumption was in Ciorești locality, which is a typical forest area for the central part of the country from which much more wood was available compared with the other two localities (and regions). We assume this level of consumption is also influenced by the availability of wood resources, which is the case of the central region in general. For comparison, the average household's annual consumption of firewood, estimated at 6.2 $m^3$ for central Moldova, almost matches the wood consumption in Suceava County of neighboring Romania with 6.1 $m^3$ [69], a county that harbors a forest area larger than the entirety of Moldova on a surface five times smaller. The average level of annual firewood consumption per the three sampled localities in our study was 5.2 $m^3$, which is slightly higher than the 4.8 $m^3$ found in the 2014 study. Both figures are higher than the NBS official estimations of 3.6 $m^3$ in the rural area and 3.5 $m^3$ at the country level for the period 2015–2016 [48]. The differences between sources regarding the annual firewood consumption may be explained by the methodological differences. However, all these sources, as well as an earlier FLEG analysis done by a local NGO in cooperation with local public authorities [47], demonstrate that nothing other than firewood consumption exceeds the officially authorized supply. All of this information raises suspicion about an existing (and probably well-handled) illegal extraction and possibly large amounts of uncountable wood that escapes reporting and control by law enforcement agencies. A high dependence on firewood, a social demand

supplied economically through illegal logging, brings forward the urgent need to promote sustainable alternative heating resources [70]. To reduce the amount of firewood used, such as through forestry–energy cooperation [21] and a reorientation towards ecological forestry [71], will certainly decrease the pressure on existing forest ecosystems by placing them in the care of 'softer management'.

In terms of how to widen access to wood resources, we looked at the case of Armenia [66] where forest-dependent households were allowed to harvest up to 8 m$^3$ of firewood per year, but few people obtained the wood directly from the forest because of their age (most people being old and unable to collect fallen wood) and it was also cheaper to buy firewood than to rent the equipment needed to harvest and transport the wood. This system may be difficult to implement and control under the conditions present in Moldova, but it might be more sensible to reduce the price of firewood, especially for those who need it most. This can be achieved through more supplies from energy plantations/crops or increased opportunities for new companies to enter the sector [72]. In addition, sustainable logging/production and improved forest management can 'relaunch' the forestry sector as, according to many sources, Moldsilva's forest entities have been assessed as having low productivity [34].

### 6.4. Factors Influencing Dependence on Forest Resources

The availability of forest products that can potentially be obtained from the forestlands is the most influential factor of forest dependence [59], with such dependence being higher in regions surrounded by forests and isolated from large cities or infrastructure [62]. The closer the community is to the forest, the more forest resources are used [22,58]. The present study confirms these considerations through the following arguments: (1) the share of forest resources used in total income is higher in households located at shorter distances from the forest, (2) households located near a richer forest resource use more firewood (as in case of Ciorești), and (3) fewer forest products are actually collected by local people if the forest is at a greater distance (case of Borceag).

The level of total income does not significantly influence the use of forest resources, especially as all households need firewood which is only available at high prices. At the same time, it is quite evident that lower-income households are dependent on the wood resource through the expenses they incur. The same trend can be identified if the quintiles distributed based on wealth are followed, so households with a more modest wealth are again those with the highest expenses. Differences can be seen in the level of products collected from the forest, so in middle-income households the quantity of products is higher. The quantity decreases toward the last quintile, due to the low value of these resources, so as the income of the locals increases, these resources are of less interest to them, and the dependence on the forest is lower [73]. At the same time, the quantity decreases further towards the first quintile, where besides the lower income, the population is also older. Other studies indicate the same point when age is another important factor, so the older the people are, the less they harvest from the forest [74,75].

Another factor determining forest dependence is the level of education. Numerous studies [22,73,74,76] suggest that the more years of education the household head/household members have, the lower the dependence on forest products is. In the case of Moldova, this aspect is less highlighted, but our analysis shows that in households with members having less education, the products used have a higher share in total income compared to those with more educated members, which means a higher level of dependence.

### 6.5. Directions for Strategies and Policies in Moldova

Evidence from our research can greatly inform decisions about sectoral policies and identify new issues for large-scale projects. In addition, this is the case of the recently launched initiative entitled the National Afforestation and Reforestation Plan (NARP), announced by the Ministry of Environment under leadership of the Presidency of Moldova [77]. The target indicators of the NARP are 100,000 hectares of land for the next 10 years

(2023–2032) split into two operational components: (a) afforestation activities mainly focused on production forests, and (b) reforestation through rehabilitation of native or close-to-nature approaches. Experts suggest [77] that achieving these indicators is only possible in two circumstances: (a)with the involvement of the private and municipal (community) sector in extension, and (b) through the proper management of existing public forests. This is not at all an easy task as Moldova never experienced such 'afforestation/reforestation' volumes in the forestry sector, for example, the increase in the area covered by forests from 1983 to 2021 was only 69.5 thousand hectares [42]. On the other hand, the increase of forest area is considered absolutely necessary [78], given that currently the forest area is only 370.7 thousand hectares or about 11.0% of the country's territory area [42], but the target is to have at least 15% forest cover [79].

Forest restoration, reforestation, or afforestation bring short-term benefits that help improve the living standards of the population, for example, with subsidies to low-income rural households for afforestation activities [80]. When social needs are identified and well quantified, new plantations/crops that meet said needs can be grown accordingly. For Moldova, forests perform essential functions for meeting people's needs in a sustainable way and tackle climate change, especially in vulnerable areas (e.g., under desertification or eroded lands). In addition to establishing forests that closely follow the natural forest type in applying afforestation/reforestation techniques and methodologies, creating energetic plantations based on fast-growing species (including exotic but tested crops) should be highly considered. Undoubtedly, the cyclical production of forests for the purpose of providing firewood would reduce the impact on natural forests and provide opportunities for their recovery [81]. Afforestation initiatives should therefore aim to consider social needs as much as possible, so that human well-being and forest cover (mainly related to plantation type and use) can coexist in a win–win balance [12].

Some of the respondents in our survey who claimed to be unskilled seasonal workers said they participated in various types of forest work. Payment for forest work is usually less attractive than for agricultural companies, and forest work is carried out over shorter periods (with a lower respective total income). Eventually, the lack of a stable workforce and incapacity to train/grow new workers are likely to lead to work being carried out in an inadequate manner. If the NTFPs harvested by local people do not influence their welfare (not to mention timber products that are difficult to afford as income), current forest management needs to be reviewed to help (and not distance) local communities and provide more employment opportunities with better paid jobs.

An effective communication with the forest dependent communities provides people with more information and a positive 'atmosphere' that helps them manage various issues [28,30,68]. The community needs to be involved in forest management by: (a) identifying their needs and ways to meet livelihoods in a sustainable way (also diversifying income sources), (b) assuring their involvement in carrying out forest institutional reforms [30,82] and creating infrastructure facilities [83], and (c) by ensuring the transparency of the legal frame where anyone can participate in improving legislation [28]. In this way, initiatives related to the sustainable management of forest ecosystems remain not only at the intention/concern stage but are also possible to achieve. Forests can become true pillars by motivating the population to stay in the country, especially in the context of the massive emigration in recent years [49]. The relationship between the rural population and forests can achieve sustainable partnerships with mutual benefits, especially within the context of addressing both poverty and forest expansion. Ecotourism can also have an important impact in alleviating poverty [80], so any of the ongoing initiatives can be encouraging; however, ecotourism should be rational and nature-friendly [31].

Besides securing revenues to state/local budgets, the forestry sector can be seen as a means of alleviating poverty in rural areas, including through energy/fruit plantations. The forestry sector can undoubtedly help meet social needs through improved livelihoods. However, it also provides opportunities to rehabilitate/restore degraded forests caused by former unsustainable practices, as nearly 80% of current Moldovan forests are of vegetative

origin [36]. This historic 'vegetative provenance' formed from grown sprouts of 2nd, 3rd, or even 4th regeneration phases are vulnerable to climate change, and thus inopportune for the modern forestry sector where forest resilience will greatly depend on specific species vitality. Any proposed reconstruction/restoration or afforestation/reforestation initiative should never jeopardize peoples' lives but raise their morale and consolidate their capacities instead. Forest expansion, including through new energetic or fruit plantations, would very much encourage rural communities and allow them to become allies in sustainable projects (e.g., tourism, food forests, bioenergy, conservation, etc.). At the end, all this will help the rural population raise more income and improve their well-being, which will optimally assist the sustainability initiative alongside it.

## 7. Conclusions

Based on our findings from a comparative analysis of the 2014 forest dependency analysis in Moldova and the present research, we can conclude the following:

Moldova's state forest authorities subtly balance intense management and the requested conservation: state-owned forest entities under the governmental Agency Moldsilva are the main players in human–forest relationships, they generate more forest income than private and community-owned forests both per household and per hectare.

Forest dependency is more than just a forest income: although local communities do not earn income from forest resources besides a small amount of 1% from their total earnings, they are considered forest dependent, and because of their restricted access to forest resources, local people must extract considerable amounts of money, reaching an average of 9.98%, from their income to substitute these forest resources.

The highest dependency is identified among lower income households: with the level of family/household expenditure for wood (namely firewood) estimated at a very high share of 18.8% from the total income, it is difficult for this group to escape poverty when they must purchase wood at such a high price.

Attractiveness of collecting NTFPs has decreased: at less than 1% from 6.6% of the total income of rural households compared to the 2014 analysis, in 2020, the average total income per person increased and covered the subsistence minimum, but the availability of the NTFPs decreased.

Availability and distance to forest resource can influence its use: the higher availability and the shorter distance (less than 2 km), the more products are accessed and used; increased levels of income and education can reduce dependence on forests.

Dependency on firewood is high: with wood for heating being the main forest product used to describe the social-economic dependency on domestic forests; however, its energetic efficiency is reduced (mainly because it is used 'wet', and that simply reduces its thermic efficiency).

A high annual consumption of firewood per household is once again demonstrated by this research with an average level of 5.2 m$^3$, which is slightly higher than the amount of 4.8 m$^3$ found in the 2014 analysis and those amounts confirmed with other studies too, supporting the notion of a higher demand than the potential supply, which should be taken into consideration by decision-makers.

Illegal logging may occur where the need for firewood is not met: there is a clear need to balance forest resource dependency and to conserve biodiversity; therefore, it is essential to appreciate local communities by providing alternative ways to tackle this socio-economic need for firewood and related wood products.

Solutions must be cross-sectoral and be enacted over a long-term: there is a need to involve the rural population in promoting forest policies that includes stakeholders, an implicitly rural population, into broader discussions.

**Author Contributions:** Conceptualization, N.T. and B.P.; methodology, N.T. and B.P.; validation, B.P., A.L. and A.F.H.; investigation, N.T.; writing—original draft preparation, N.T.; writing—review and editing, B.P., A.L. and A.F.H.; supervision, B.P. All authors have read and agreed to the published version of the manuscript.

**Funding:** This research received no external funding.

**Informed Consent Statement:** Informed consent was obtained from all subjects involved in this study.

**Acknowledgments:** Special thanks are addressed to all members of the studied communities who gladly took part in our interviews. We are grateful to the staff of the Forest Research and Management Institute (ICAS Chișinău) and other subunits of Agency Moldsilva for their various froms of help and advice during our study.

**Conflicts of Interest:** The authors declare no conflict of interest.

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
