# Peer review of "Forest Dependence of Rural Communities in the Republic of Moldova"

_forests, doi:10.3390/f13060954_

Round 1
Reviewer 1 Report
I congratulate authors on terrific paper. Detailed comments are provided in the pdf document. Here are some general notes
- The paper is interesting and the topic is appropriate and cutting edge
- Additional explanations are needed in the Methods, so the readers can better understand it.
- Figures and map are illustrative and it is good that you added them, because they help readers to better understand the results

Author Response
Reviewer 1 (modifications are made in Blue in the revised manuscript)
C1. Additional explanations are needed in the Methods, so the readers can better understand it.
Answer: Thank you for your comments and suggestions.
Related to the 1st comment:
In what sense – the 3rd household? 3rd in left or right from that house, 3rd in the phone book, 3rd in the list from local self-government. Please describe:
- first household from the entrance to the locality were approached, from this house were approached the third household from it, either the right or on the left side of the road. Changes were applied in the text in blue color.
About the 2nd comment:
Please provide further explanation about the criteria used. I understand the point, but seems like a very subjective approach, i.e. if somebody wanted to repeat this research in some other country - what to do?
Indeed the ”snowball” sampling wasn`t well described. The changes were applied as it follows:
Besides the standard questionnaire, for 10 representative persons in each village who occupy higher social or technical positions, questionnaire 2 was used in order to collect qualitative data on the most important livelihood products as well as more information on infrastructure, markets and prices of products. For this cause, three persons who knew better the general situation in each village were approached, town hall employee, social worker, and forester. After interviewing these persons, they were asked to give contacts of other people who can know best the situation in that village.
Regarding the 3rd comment:
I don`t understand what this means. Were they asked to give the score from 1 to 3? Please explain here and for the rest of the results.
- in the tables 1 and 2 is written that the answers were rated on a scale of 1 to 3. For a better understanding, additional explanation was given. Changes can be found in the improved version of the manuscript in tables 1 and 2 in blue color.
Thank you also for the applied correction in the conclusions section.

Reviewer 2 Report
The manuscript gives an important insight into the problem of forest dependency of the local communities in Moldova. The study is a follow-up of the previous work done in 2014. No doubt, the paper is an important contribution to the literature, however, it must be revised to follow the standards of a high-quality journal.
1. The title of the paper is not concise and must be reformulated. The term “forest dependence“ needs clarification. Maybe adding the “local communites”? Is the term “among“ appropriate in this case?
2. The abstract is short and would benefit from some extension. The parentheses-included comments (such as “still facing poverty challenges”) need to be incorporated in the text.
3. Some illustrative material would be nice to see in section 2. The authors give a lot of figures characterizing the national context, but they are not clear when reading superficially. It does not attract the reader.
4. Figure 1 is not readable at all. Too many unnecessary details and very hard to understand where the three study areas are located.
5. The conclusions are not supported with appropriate figures. The same goes for abstract. It is important to repeat the key interview-based quantitative facts that strenghten the argumentation of the importance of forest dependency of the local communities.
Author Response
Reviewer 2 (modifications are made in Green in the revised manuscript)
C1. The title of the paper is not concise and must be reformulated. The term “forest dependence“ needs clarification. Maybe adding the “local communites”? Is the term “among“ appropriate in this case?
Answer: Thank you for the comment and the suggestions.
The title is re-written and shown in Green color in the improved version of the manuscript: Forest Dependence of Rural Communities in the Republic of Moldova.
C2. The abstract is short and would benefit from some extension. The parentheses-included comments (such as “still facing poverty challenges”) need to be incorporated in the text.
Answer: Thank you for your suggestion.
The abstract has been extended, and the parentheses have been incorporated in the text. Added information is shown in green color in the improved version of the manuscript.
C3. Some illustrative material would be nice to see in section 2. The authors give a lot of figures characterizing the national context, but they are not clear when reading superficially. It does not attract the reader.
Answer: Thank you for your comments and suggestion.
We agree that for the reader this section is not very attractive, so we added some illustrative material.
We have implemented this suggestion by introducing two new figures (Figure 1 and Figure 2) that can be found in the improved version of the manuscript with their names colored in Green. First figure describes the relationship between monthly average value of the total income and the subsistence minimum, and the second figure shows the share of different sources of income in the total disposable income of the rural population in the Republic of Moldova.
C4. Figure 1 is not readable at all. Too many unnecessary details and very hard to understand where the three study areas are located.
Answer: Thank you so much for your comment.
The unnecessary details were excluded from the figure. The three snapshots were introduced initially to demonstrate the presence of forest resources near sampled villages. Those snapshots were indeed unnecessary as in the general map, the distribution of forest around the villages already can be noticed. The modified figure was introduced in the improved version of the manuscript (as Figure 3 now).
C5. The conclusions are not supported with appropriate figures. The same goes for abstract. It is important to repeat the key interview-based quantitative facts that strengthen the argumentation of the importance of forest dependency of the local communities.
Answer: Thank you for your comments.
Regarding your suggestions, we added some quantitative facts in the conclusions section and for abstract we integrated additional information consistent with addressed suggestions from 2nd comment about extending the abstract. Added information is shown in green color in the improved version of the manuscript.

Round 2
Reviewer 2 Report
Authors made a substantial progress with the manuscript. Good luck.